# Enantioselective Recognition of Lysine and Phenylalanine Using an Imidazole Salt-Type Fluorescent Probe Based on H_8_-BINOL

**DOI:** 10.3390/molecules27238470

**Published:** 2022-12-02

**Authors:** Zhaoqin Wei, Shi Tang, Xiaoxia Sun, Yu Hu

**Affiliations:** 1Jiangxi Key Laboratory of Organic Chemistry, Jiangxi Science and Technology Normal University, Nanchang 330013, China; 2College of Chemistry, Nanchang University, Nanchang 330031, China

**Keywords:** H_8_-BINOL, fluorescent probe, enantioselective recognition

## Abstract

An imidazole bromide fluorescent probe (*R*)-**1** based on chiral H_8_-BINOL was synthesized with a high yield; it was found to have good enantioselective recognition of lysine and phenylalanine using fluorescence analysis. When L-lysine was recognized, the enantioselective fluorescence enhancement ratio was 2.7 (ef = I_L_ − I_0_/I_D_ − I_0_, ef = 2.7, 20 eq Lys); as the amount of L-Lys increased, a distinct red shift was observed (the wavelength varied by 55.6 nm, 0–100 eq L-Lys), whereas D-Lys had a minimal red shift. The generation of this red shift phenomenon was probably due to the ICT effect; the probe’s intramolecular charge transfer was affected after (*R*)-**1** bound to L-Lys, and this charge transfer was enhanced, leading to a red shift in fluorescence. In addition to the enantioselective recognition of lysine, phenylalanine was also recognized; the enantioselective fluorescence enhancement ratio was 5.1 (ef = I_L_ − I_0_/I_D_ − I_0_, ef = 5.1, 20 eq Phe).

## 1. Introduction

In the past few years, enantioselective fluorescent probes have become a research hotspot and have been widely used for the enantioselective recognition of some amino acids, amino alcohols, and chiral amines [1,2,3]; these chiral molecules’ roles are important in organisms and chemical research. In the history of chemical research, many methods have been used to recognize chiral molecules; fluorescence analysis was widely used in enantioselective recognition due to its convenience, high sensitivity, fast and high-throughput analysis, and biological imaging [4]. Chiral amino acids are widespread in nature and play important roles in chemical and biological systems; for example, some amino acids can be used for asymmetric metal organocatalysis [5]. Naturally occurring amino acids exist primarily in L-amino acids, the structural units of proteins. However, in recent years, more and more D-amino acids with a wide variety of biological functions have been found in plants, organisms, and foods. In humans, a lack of phenylalanine may cause phenylketonuria; in children, a lack of lysine may cause developmental retardation, anemia, and other symptoms. Previous research has reported that Alzheimer’s disease has a close relationship with D-aspartate and D-serine [6]. Therefore, the enantioselective recognition of amino acids has become increasingly important, and many studies have reported achieving a high enantioselective recognition of opposite amino acids by designing different classes of fluorescent probes [7,8,9,10].

Over the past decade, many studies have used a series of fluorescent probes based on BINOL, showing a high enantioselective recognition of amino acids. BINOL has been widely used to synthesize chiral fluorescent probes due to its two naphthalene rings, good rigidity, and C2-axis chirality [11,12,13]. However, there are few studies based on H_8_-BINOL, which is primarily involved in asymmetric catalysis and asymmetric synthesis, and few reports of the use of H_8_-BINOL in chiral fluorescent probes [14,15,16]. H_8_-BINOL, a partially hydrogenated derivative of BINOL, due to SP3 carbon atoms hybridization, an increased volume, and an increased electron density on the aromatic ring, significantly differs from BINOL [17].



Therefore, H_8_-BINOL provided the fluorescent probe’s chiral environment and the origin of its fluorescence response; the diimidazole moiety was led into H_8_-BINOL’s third position, providing possible binding sites for the chiral substrate. The synthesis process is shown in Figure 1 below. The fluorescence response of (*R*)-**1** and (*S*)-**1** to 15 amino acids in an aqueous solution was explored using the fluorescence analysis method. It was found that (*R*)-**1** can achieve the enantio-selective recognition of lysine and phenylalanine. (*R*)-**1** had an obvious fluorescence response after the addition of L-lysine and L-phenylalanine, accompanied by significant red shift and blue shift phenomena, whereas D-lysine and D-phenylalanine did not. Upon the addition of 20 eq L-lysine, the enantiomer fluorescence enhancement ratio was 2.7 (ef = I_L_ − I_0_/I_D_ − I_0_, ef = 2.7), and the phenylalanine was 5.1 (ef = I_L_ − I_0_/I_D_ − I_0_, ef = 5.1).

## 2. Results and Discussion

The fluorescent probe (*R*)-**1** was synthesized using the steps outlined in Figure 1. The fluorescence responses of the fluorescent probe (*R*)-**1** to fifteen amino acids (D/L-Phe, D/L-Ala, D/L-Met, D/L-Pro, D/L-Glu, D/L-Gln, D/L-Arg, D/L-Lys, D/L-Leu, D/L-Ser, D/L-Thr, D/L-Asn, D/L-Asp, D/L-Val, and D/L-His) were investigated in an aqueous solution using a Hitachi F-7100 fluorescence spectrophotometer. (*R*)-**1** and amino acids were dissolved in deionized water, with an (*R*)-**1** concentration of 2.0 × 10^−5^ M and amino acid concentrations of 0.1 M. When 20 eq amino acid was added to the probe (*R*)-**1**, (*R*)-**1** exhibited different fluorescence responses to L-Lys and L-Phe compared to other amino acids at λexc = 280 nm (Figure 1). The fluorescence intensity was significantly enhanced, and the L-Lys wavelength moved in the long-wave direction, whereas the D-Lysine wavelength was almost unchanged. The L-Phe wavelength shifted slightly in the short-wave direction, whereas the D-phenylalanine wavelength did not change. This series of changes suggested that (*R*)-**1** can enantioselectively recognize lysine and phenylalanine.

### 2.1. Fluorescence Experiments of (R)-***1*** on Lysine

Both (*R*)-**1** and amino acids were prepared in deionized water; the (*R*)-**1** concentration was 2.0 × 10^−5^ M, and the lysine concentration was 0.1 M (2 mL in volume). Next, 100 eq D/L-lysine was added to (*R*)-**1** to observe the fluorescence response. At λexc = 280 nm, L-Lys’ fluorescence intensity was enhanced, and the wavelength was accompanied by a slight red shift phenomenon, whereas D-Lys’ fluorescence intensity increased only slightly, with almost no change in the wavelength. This indicated that (*R*)-**1**’s fluorescence responses to the two configurations of lysine are different and can be differentiated to some extent (Figure 2).

To further explore using (*R*)-**1** to identify Lys, we performed fluorescence titration experiments on D-Lys and L-Lys separately, with an (*R*)-**1** concentration of 2.0 × 10^−5^ M and a lysine concentration of 0.1 M (volume 2 mL); both were prepared in deionized water, added from 0 to 100 eq. During L-Lys titration at λexc = 280 nm (Figure 3a), as the L-Lys amounts increased (from 0 to 100 eq), the fluorescence intensity significantly and steadily increased by 2.3 times (from 1316 to 3000), and the wavelength moved toward the long wave and red-shifted by 55.6 nm from 348.6 nm to 404.2 nm, with a large amplitude. During the titration of D-Lys (Figure 3b), the fluorescence intensity also increased as the D-Lys amounts increased (from 0 to 100 eq); however, the increase was smaller than that of L-Lys, and the wavelength range (from 349.4 nm to 355.8 nm) was small. Figure 3c illustrates that the fluorescence intensity rose relatively steadily as the amount of lysine increased and reached a good linear relationship of R = 0.9982.

(*R*)-**1**’s stability in an aqueous solution was tested; at λexc = 280 nm, (*R*)-**1**’s fluorescence intensity increased within 30 min, but its wavelength did not change (Appendix A). Therefore, the small increase in D-Lys’ fluorescence intensity may have been influenced by (*R*)-**1**. In terms of fluorescence intensity, (*R*)-**1**’ recognition effect for Lys was not obvious; however, with an equivalent increase in Lys, a change in the wavelength between the two Lys configurations showed that (*R*)-**1** can achieve a good recognition effect for L-lysine.

To investigate lysine’s enantiomer composition, the fluorescence response of (*R*)-**1** and lysine at different ee values (ee% = [L] − [D]/[L] + [D]) was explored at λexc = 280 nm; its fluorescence intensity changed when lysine was mixed with (*R*)-**1** at different ee values. Lysine’s enantiomer composition is presented in Figure 4.

In conclusion, to explore the recognition mechanism of (*R*)-**1** and lysine, we conducted ^1^H NMR spectroscopic studies. (*R*)-**1** was dissolved in DMSO, and L-Lys was dissolved in D_2_O, adding 1.0 eq L-Lys to 2.0 eq into the (*R*)-**1** (V_DMSO_/V_D2O_ = 10/1) to observe changes in the ^1^H NMR hydrogen spectra. According to ^1^H NMR spectra (Figure 5), the signal peak disappeared for (*R*)-**1**’s Ha at 9.78 ppm, and its chemical shift of Hb, Hc, and Hd moved slightly towards the high-filed aera, Hb moved from 8.19 to 7.94 ppm, Hc moved from 7.81 to 7.68 ppm, and Hd moved from 6.84 to 6.67 ppm, with range variations of 0.25 ppm, 0.13 ppm, and 0.13 ppm respectively. H’s chemical shifts in the other group shown changed slightly; however, H’s signal peak in the benzene ring did not change. This series of changes indicates that there were indeed some changes between (*R*)-**1** and L-Lys after the addition of L-Lys and that these changes occurred on the imidazole group. Therefore, fluorescent probe (*R*)-**1**’s recognition group might be the molecule’s imidazole group.

The structural analysis of (*R*)-**1** indicated that H_8_-BINOL served as the probe’s fluorophore and that the imidazole group served as the recognition group. (*R*)-**1**, as the host molecule, interacted with the guest molecule L-Lys, this might be due to the electron-donating effect of lysine’s amino group in the end group and the recognition group’s electron deficiency, which, together, caused the probe to react with L-Lys. After the interaction of the host and guest molecules, the imidazole group’s electron absorption capacity was reduced, which affected the charge transfer within the probe. This process enhanced the intramolecular charge transfer (ICT) [18,19,20,21,22], resulting in a red shift of fluorescence and an enhancement of the fluorescence intensity. As seen in Figure 3a, during lysine titration, the wavelength shifted from 348.6 nm to 404.2 nm, showing an increase of 55.6 nm. The fluorescence intensity increased by 2.3 times from 1316 to 3000; however, during D-Lys titration, there was no red shift, and its fluorescence intensity only ranged from 1354 to 1877. Most ICT fluorescent probes have an obvious shift in spectra after binding to guest molecules, but the change in fluorescence intensity was not obvious. The lysine enantioselective recognition effect was identified and characterized by the obvious red shift phenomenon and by the changes in the fluorescence intensity.



### 2.2. Fluorescence Experiments of (R)-***1*** on Phenylalanine



A preliminary exploration of (*R*)-**1**’s capability to recognize phenylalanine was undertaken. As shown in Figure 6a, adding 100 eq D/L-phenylalanine to (*R*)-**1,** respectively, at λexc = 280 nm resulted in a relatively obvious fluorescence response to L-Phe, with a significant enhancement of fluorescence intensity and a small offset in wavelength, whereas D-Phe had almost no change in fluorescence intensity and wavelength. According to Figure 6b, as the L-Phe amount gradually increased (from 0 to 100 eq), the fluorescence intensity steadily increased, and its wavelength experienced a slight blue shift in the short-wave direction, whereas D-Phe’s fluorescence intensity remained almost unchanged, and its wavelength did not change (Figure 6c). L-Phe’s fluorescence intensity remained relatively constant with an increasing equivalent (from 0 to 100 eq) and showed a good linear relationship of R = 0.9979, as shown in Figure 6d.

The enantiomeric composition of phenylalanine was also studied; (*R*)-**1** was mixed with phenylalanine at different ee values at λexc = 280 nm to observe changes in the fluorescence response. The enantiomeric composition of phenylalanine was determined, as shown in Figure 7.

## 3. Conclusions

In conclusion, an imidazole bromine salt fluorescent probe was designed and synthesized based on H_8_-BINOL. The probe was found to have a good enantioselective recognition effect of lysine and phenylalanine using the fluorescence analysis; the enantiomer composition of amino acids could also be determined. The enantioselective recognition of phenylalanine and lysine by (*S*)-**1** (enantiomer of (*R*)-**1**) was also explored (Appendix A); both were found to recognize the same configuration of amino acids. Unlike the traditional enantioselective fluorescence probe, there was a mirror relationship between different configurations for chiral molecule configuration recognition. The recognition mechanism of (*R*)-**1** to lysine can be referred to as an ICT (intramolecular charge transfer) mechanism; when (*R*)-**1**’s recognition group bound to lysine, the intramolecular charge transfer occurred, and this charge transfer was enhanced, resulting in a fluorescence red shift. This allowed for the enantioselective recognition of lysine and phenylalanine not only via a significant change in the fluorescence intensity but also by a significant change in the wavelength after the addition of amino acids. Amino acids are essential to the human body; they are important components of proteins. Therefore, the enantioselective recognition of amino acids has important significance; the enantioselective fluorescence imaging of amino acids in human biological systems using chiral fluorescence probes could assist in the diagnosis and assessment of some diseases.

## 4. Methods and Materials

The experimental reagents and chemicals used were purchased. Methanol and acetonitrile were further treated with water removal; other reagents were directly used without further purification. A Bruker AM-400WB spectrometer measured ^1^H NMR and ^13^C NMR, and an X-4 measuring point tester measured melting points. Optical rotation was performed using a Rudolph AUTOPOL IV automatic polarimeter in chromatographic methanol. Fluorescence experiments were conducted using a Hitachi F-7100 Fluorescence spectrophotometer. Elemental analyses were carried out a Vario EL/MACRO elemental analyzer to obtain C, H, and N amounts in molecules.

### 4.1. Experimental

The synthesis steps of all compounds, ^1^H NMR, and ^13^C NMR are presented in the Appendix A. Compound **1** and Compound **2** were synthesized with reference to a previous study [17].

Step 1: Synthesis and Characterization of Compound **1**

Under argon, NaH (1.63 g, 67.94 mmol) was added to a round bottom flask, and 8 mL anhydrous tetrahydrofuran was added to dissolve the NaH and cool the system to 0 °C. Next, (*R)*-H_8_-BINOL (5.0 g, 16.98 mmol) was dissolved by slowly adding ultra-dry THF and stirring for 10 min; then, bromomethyl methyl ether (3.2 mL, 39.06 mmol) was added, and the mixture was continuously stirred during the reaction at 0 °C for 8 h under an argon atmosphere. When TLC detection showed that the reaction was complete, the mixture was quenched with ice water and extracted with ethyl acetate three times. The organic phase was washed with saturated NaCl solution and then dried for 30 min using anhydrous MgSO_4_. The solvent was removed and purified by column chromatography on silica gel eluted with petroleum ether/ethyl acetate (35:1); a sticky dark-green solid product (6.35 g) with a 97% yield was obtained. ^1^H NMR (400 MHz, Chloroform-d) *δ* 7.13–6.89 (m, 2H), 5.14–4.83 (m, 2H), 3.28 (s, 3H), 2.77 (s, 2H), 2.20 (d, J = 61.7 Hz, 2H), 1.69 (d, J = 33.5 Hz, 4H). ^13^C NMR (101 MHz, Chloroform-d) *δ* 152.38, 136.92, 131.07, 128.92, 127.29, 112.95, 94.99, 29.57, 27.41, 23.36, 23.25, 23.25.

Step 2: Synthesis and Characterization of Compound **2**

Under argon, the product (3.19 g, 8.3 mmol) from step 1 was added to a round-bottom flask and dissolved using 20 mL anhydrous tetrahydrofuran (THF). Next, 10 mL n-BuLi (2.5 M, 25.0 mmol) was slowly added under 0 °C; the reaction mixture’s color gradually changed from colorless to dark brown. The reaction mixture was continuously stirred for 1 h. While adding anhydrous N and N-dimethylformamide (DMF, 1.92 mL, 25.0 mmol), its color changed from dark brown to light yellow; then, a stirring reaction occurred for 1 h. TLC detection showed that the reaction was complete; the mixture was quenched three times using a saturated NH_4_Cl solution and extracted three times using dichloromethane (CH_2_Cl_2_). The organic phase was washed with saturated sodium NaCl solution and then dried for 30 min using anhydrous MgSO_4_. After that, the solvent was removed using petroleum ether/ethyl acetate (20:1) for column chromatography separation. A light-yellow solid (1.48 g) with a 43% yield was obtained. ^1^H NMR (400 MHz, Chloroform-d) *δ* 10 (s, 1H), 7.61 (s, 1H), 7.13–6.87 (m, 2H), 5.24–4.88 (m, 2H), 4.62 (d, J = 5.7 Hz, 2H), 3.32 (s, 3H), 3.15 (s, 3H), 2.93–2.62 (m, 4H), 2.37 (s, 2H), 2.19 (s, 2H), 1.71 (d, J = 22.6 Hz, 8H). ^13^C NMR (101 MHz, Chloroform-d) *δ* 191.00, 155.50, 152.59, 145.50, 134.19, 132.09, 131.29, 129.75, 128.10, 127.69, 125.29, 111.84, 100.05, 94.74, 57.19, 56.07, 29.61, 29.53, 28.12, 27.51, 23.25, 23.15.

Step 3: Synthesis and Characterization of Compound **3**

Compound **2** (1.48 g, 3.6 mmol) from step 2 was dissolved using 4 mL of anhydrous tetrahydrofuran (THF) in the reaction bottle. Subsequently, 6 mL of dried methanol was added, and the contents were stirred and dissolved in an ice water bath. Next, NaBH_4_ (0.54 g, 14.43 mmol) was added, and the reaction mixture was stirred and sustained for 30 min. When TLC detection showed that the reaction was finished, the mixture was quenched with ice water, extracted three times using ethyl acetate, washed with saturated NaCl solution, dried with anhydrous MgSO_4_, and separated using petroleum ether/ethyl acetate (5:1) column chromatography to obtain a white solid (1.45 g) with an 86.3% yield. ^1^H NMR (400 MHz, Chloroform-d) δ 7.11 (s, 1H), 7.09–6.97 (m, 2H), 5.13–5.02 (m, 2H), 4.75 (d, J = 6.2 Hz, 2H), 4.49 (d, J = 5.9 Hz, 2H), 3.35 (d, J = 6.9 Hz, 6H), 2.81 (s, 4H), 2.30 (s, 2H), 2.11 (s, 1H), 2.00 (s, 1H), 1.70 (d, J = 36.4 Hz, 8H). ^13^C NMR (101 MHz, Chloroform-d) δ 152.52, 151.71, 137.25, 137.08, 133.77, 131.79, 131.14, 129.99, 129.34, 126.35, 111.67, 99.06, 94.63, 61.65, 56.92, 56.05, 29.65, 29.54, 27.47, 27.36, 23.27, 23.23, 23.16, 23.10.

Step 4: Synthesis and Characterization of Compound **4**

Under the protection of argon, Compound **3** (0.6 g, 1.45 mmol) from step 3 was dissolved in anhydrous toluene and stirred in an ice water bath for 10 min. Later, methylsulfonyl chloride (0.22 mL, 2.9 mmol) was slowly added over 10 min; then, triethylamine (0.6 mL, 4.36 mmol) was slowly dripped into the reaction system and stirred for 2 h. Next, LiBr (0.25 g, 2.9 mmol), dissolved using N,N-dimethylformamide, was added to the reaction, and the reaction was completed for 1 h. The resulting mixture was quenched in H_2_O (10 mL) and extracted using EtOAc (3 × 20 mL), and then the combined organic phase was washed with saturated NaCl solution and dried using anhydrous MgSO_4_. Further purification using column chromatography on silica gel was conducted by gradient elution with 4–5% ethyl acetate in petroleum ether, which obtained Compound **4** as a white solid (0.6 g) with an 88% yield. ^1^H NMR (400 MHz, Chloroform-d) δ 7.18 (s, 1H), 7.09–6.94 (m, 2H), 5.14–5.00 (m, 2H), 4.80 (d, J = 5.1 Hz, 1H), 4.73 (s, 2H), 4.55 (d, J = 5.2 Hz, 1H), 3.34 (s, 3H), 3.14 (s, 3H), 2.78 (d, J = 17.7 Hz, 4H), 2.37 (s, 2H), 2.10 (d, J = 28.1 Hz, 2H), 1.70 (d, J = 26.6 Hz, 8H). ^13^C NMR (101 MHz, Chloroform-d) δ 152.66, 138.28, 137.18, 133.76, 131.51, 131.10, 130.51, 129.39, 128.35, 126.16, 111.72, 99.13, 94.64, 56.73, 56.00, 29.62, 29.55, 27.43, 23.28, 23.17, 23.01.

#### 4.1.1. Synthesis and Characterization (*R*)-**1**

Accurately weighed Compound **4** (560 mg, 1.18 mmol), which was synthesized in step 4, and N,N-diimidazoylmethane (87 mg, 0.59 mmol) were dissolved in 15 mL anhydrous acetonitrile at room temperature after being heated at reflux for 48 h in an oil bath at 80 °C. After the reaction completed, the mixture cooled to room temperature, and then the resulting solution was concentrated under reduced pressure to remove the solvent, which was further purified by column chromatography on silica gel eluted with CH_2_Cl_2_/CH_3_OH (15:1) to obtain (*R*)-**1** as a white solid (480 mg) with a 74.1% yield. ^1^H NMR (400 MHz, DMSO-d_6_) δ 9.78 (s, 1H), 8.19 (s, 1H), 7.81 (s, 1H), 7.06 (t, J = 4.1 Hz, 2H), 6.96 (d, J = 8.5 Hz, 1H), 6.84 (s, 1H), 5.46 (q, J = 14.7 Hz, 2H), 5.18–4.93 (m, 2H), 4.62 (d, J = 5.4 Hz, 1H), 4.45 (d, J = 5.4 Hz, 1H), 3.22 (s, 3H), 3.03 (s, 3H), 2.73–2.67 (m, 4H), 2.21 (s, 2H), 2.00 (s, 2H), 1.65 (s, 8H). ^13^C NMR (101 MHz, DMSO-d_6_) δ 152.21, 150.71, 138.50, 138.30, 136.25, 133.86, 131.48, 130.53, 129.92, 129.72, 125.19, 124.73, 123.52, 122.84, 111.78, 98.79, 94.19, 58.38, 56.81, 55.84, 48.99, 29.25, 29.06, 27.17, 27.11, 22.92, 22.71, 22.63. MS-ESI m/z: [M-Br]^+^ calcd for C_57_H_70_BrN_4_O_8_^+^ 1017.4352; found 1017.7103; [M-Br-OCH_3_] calcd for C_56_H_67_BrN_4_O_7_ 986.4193; found 986.7321; [M-2Br-CH_3_]+ calcd for C_56_H_67_N_4_O_8_^+^ 923.4953; found 923.5130. M.p.29–30 °C. ${\left[ɑ\right]}_{D}^{25}$ 37 (c = 1, CH_3_OH). Elemental analysis results: C 62.38% (calculated value 62.29%); H 6.37% (calculated value 6.42%); N 5.21% (calculated value 5.1%).

#### 4.1.2. Synthesis and Characterization (*S*)-**1**

Using the same procedure as that detailed above, (*S*)-**1** was obtained as a white solid (180 mg) with a 55% yield. ^1^H NMR (400 MHz, DMSO-*d*_6_) δ 9.70 (s, 1H), 8.14 (s, 1H), 7.81 (s, 1H), 7.00 (d, *J* = 28.2 Hz, 3H), 6.78 (s, 1H), 5.59–5.33 (m, 2H), 5.15–4.94 (m, 2H), 4.61 (s, 1H), 4.45 (s, 1H), 3.21 (s, 3H), 3.02 (s, 3H), 2.69 (s, 4H), 2.21 (s, 2H), 1.99 (s, 2H), 1.63 (s, 8H). ^13^C NMR (101 MHz, DMSO-*d*_6_) *δ* 152.21, 138.50, 138.33, 136.25, 133.83, 131.46, 130.53, 129.98, 129.92, 129.71, 125.20, 124.72, 123.48, 122.85, 111.78, 98.78, 94.20, 58.34, 56.80, 55.83, 48.99, 29.24, 29.05, 27.16, 27.10, 22.94, 22.70, 22.62. MS-ESI m/z: [M-Br]^+^ calcd for C_57_H_70_BrN_4_O_8_^+^ 1017.4352; found 1017.7119; [M-Br-OCH_3_] calcd for C_56_H_67_BrN_4_O_7_ 986.4193; found 986.7358; [M-2Br-CH_3_]^+^ calcd for C_56_H_67_N_4_O_8_^+^ 923.4953; found 923.5145. M.p.28–30 °C. ${\left[ɑ\right]}_{D}^{25}$-37 (c = 1, CH_3_OH). Elemental analysis results: C 62.21% (calculated value 62.29%); H 6.50% (calculated value 6.42%); N 5.19% (calculated value 5.1%).

## Data Availability

Not applicable.

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
