# Peer review of "Enantioselective Recognition of Lysine and Phenylalanine Using an Imidazole Salt-Type Fluorescent Probe Based on H_8_-BINOL"

_molecules, 2022, doi:10.3390/molecules27238470_

Round 1

Reviewer 1 Report

The authors proposed a new probe for the detection of lysine and phenylalanine using an imidazole salt-type fluorescent probe based on H8-BINOL.

The aim of the research is clear but experiments are not generally well designed.

The fluorescence responses after adding L-Lys and L-Phe to the probe solution are not significant. It seems that the authors ignored the fact that lysine shows fluorescence in a similar range of fluorescence as shown in the manuscript and assigned for the reaction with the probe. This could be readily tested by recording the fluorescence spectra of lysine under the same conditions (in the absence of the probe). I am convinced that the results will be the same as those obtained in the presence of the probe.

1H NMR titration experimental data do not prove the proposed reaction mechanism of the probe with L-Lys. These data only show a change of the Ha-derived peak and this is due to the change the pH of the probe solution after the addition of a large excess of lysine.

Thus, I do not think the presented manuscript is not suitable for publication in Molecules journal.

Author Response

Dear Referee

I am very grateful to your comments for the manuscript. According with your advice, we amended the relevant part in manuscript. Some of your questions were answered below. Please see the attachment about Figures 1-4.

Comment 1:

The fluorescence responses after adding L-Lys and L-Phe to the probe solution are not significant.

Apply:

Although the fluorescence response after adding 20 eq L-Lys and L-Phe to the probe was not obvious, the fluorescence response of adding D-Lys and D-Phe was different from L-Lys and L-Phe. What we research in our manuscript is enantioselective recognition, the differences between fluorescence response of the probe to two configurations of amino acids. The fluorescence intensity of the probe to L-Lys increased greatly and had a red shift phenomenon through the titration experiments, but the fluorescence intensity was only slightly enhanced and no red shift for D-Lys, these shown that the fluorescence response of the probe to two configuration of lysine was different. The fluorescence response of the probe to L-Phe and D-Phe were also different, this can distinguish the two configurations of phenylalanine. Figure 1 and 2 was in the attachment.

Comment 2:

Lysine shows fluorescence in a similar range of as shown manuscript; tested by recording the fluorescence spectra of lysine under the same conditions.

Apply:

We have done a fluorescence experiments about L-Lys without the probe under the same conditions, it was found that the fluorescence of L-Lys at 2.0*10-5 M was quietly weak, while titrated to 100 eq, the fluorescence intensity was already increased to above 5000 and shown a significant enhancement, the maximum absorption wavelength of L-Lys in the fluorescence spectrum was located in 447.0 nm, while the fluorescence intensity enhancement was not obvious when 100 eq L-Lys was added into the probe, and the maximum wavelength was only in the 404.2 nm, which was not reached the maximum absorption wavelength of L-Lys, these indicates that the enhancement and red shift of fluorescence of the probe on lysine was not attributed to L-Lys itself. Please see the Figure 3 on the attachment.

Comment 3

1H NMR titration experimental data do not prove the proposed reaction mechanism of the probe with L-Lys, the Ha-derived peak and this due to the change the pH of the probe solution after the addition of a large excess of lysine.

Apply:

In order to verify whether the 1H NMR titration experiment was only affected by the pH of lysine, same equivalent of D-Lys was added to perform the 1H NMR titration experiment on the probe. As a pair of enantiomers, D-Lys and L-Lys have the same physical properties, and the PH of D-Lys and L-Lys at the same molar concentration should also be the same. The equivalent D-Lys and L-Lys are dropped into the probe of the same concentration. If it is only the influence of PH, the NMR spectra of the two experiments should be coincident. However, the two 1H NMR spectrum does not completely coincide, which indicates that the probe effect on L-Lys is not only by the change the pH of the probe solution. Please see the Figure 4 on the attachment.

Reviewer 2 Report

The manuscript by Wei et al. entitled “Enantioselective recognition of lysine and phenylalanine by an imidazole salt-type fluorescent probe based on H8-BINOL” describes the development of a fluorescent probe for the fluorometric differentiation of enantiomeric amino acids. The concept is quite exciting and could be of interest to a broader audience.  

Although it was claimed that the fluorescence intensity of L-Lys was significantly enhanced from 1196 to 1540 a.u. but it’s not a very significant enhancement and also the slight red shifting of wavelength of 18.8 nm is not quite significant.

All the synthesized compounds are characterized properly; all the experiments have been done carefully.  

In the final probe both the phenolic –OH groups are in their –MOM-protected form, is there any specific reason for keeping those –OH in the protected state?

Authors may investigate the sensing properties of the deprotected probe also. 

Author Response

Dear Referee

I am very grateful to your comments for the manuscript. According with your advice, we amended the relevant part in manuscript. Some of your questions were answered below. Please see the attachment.

Round 2

Reviewer 1 Report

The manuscript can be accepted in present form.

Author Response

Thank you very much.